# UNDERSTANDING ADDITION AND SUBTRACTION IN TRANSFORMERS

## ABSTRACT

Transformers are widely deployed in large language models (LLMs), yet most models still fail on basic arithmetic tasks such as multidigit addition. In contrast, we show that small transformers trained from scratch can solve n-digit addition and subtraction with 99.999% accuracy. Building directly on prior work that uncovered addition circuits, we extend the analysis to subtraction and present a unified mechanistic account based on cascading carry and borrow circuits. Using a suite of 49 trained models, we apply systematic ablations and node-level constraints to validate the learned mechanisms, and release a reproducible interpretability toolkit for studying arithmetic circuits. Finally, surveying 180 publicly available LLMs, we find that only 7% can reliably perform addition, underscoring the gap between specialized small models and general-purpose LLMs. Our results show that arithmetic can be implemented exactly by tiny transformers, offering a tractable case study for mechanistic interpretability and a cautionary contrast with the persistent arithmetic failures of much larger models.

## 1 INTRODUCTION

Large language models (LLMs) achieve remarkable performance across diverse tasks, but still struggle with basic arithmetic. In a survey of 180 LLMs ranging from 1B to 405B parameters, 7% can reliably perform addition. The few that succeed typically rely on external tools, suggesting that the underlying models have not truly learned arithmetic (See App. E). In contrast, we show that transformers trained from scratch can master both addition and subtraction with near-perfect accuracy - without using external tools.

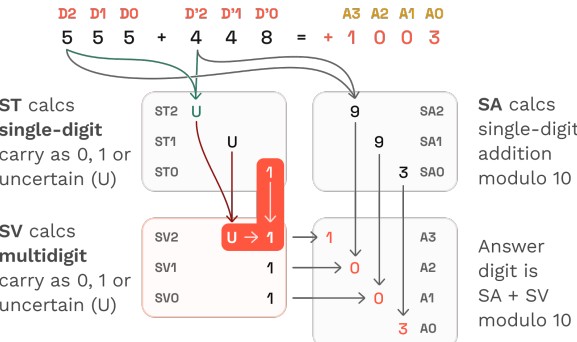

Figure 1: Our n-digit addition algorithm is mathematically sound. It uses 4 features: It calculates single-digit carry-one values ($ST_n$), combining them into multidigit carry-one values ($SV_n$). Any $SV_n$ uncertain (U) values are refined to 0 or 1 over tokens (highlighted). By the "+" token, $SV2$ gives the A3 value as 0 or 1. The other answer digits $A_n$ are calculated from $SA_n$ and $SV_{n-1}$ values.

We construct (effectively infinite) synthetic training data enriched with the hardest edge cases, and train 49 small transformer models (2–3 layers, 3–4 attention heads) on 5 to 12-digit arithmetic. These models converge rapidly, reaching >99.999% accuracy. They succeed on the cascading carry one and borrow one cases where LLMs fail e.g. 555555555 + 444444448 = 1000000003. Our largest model

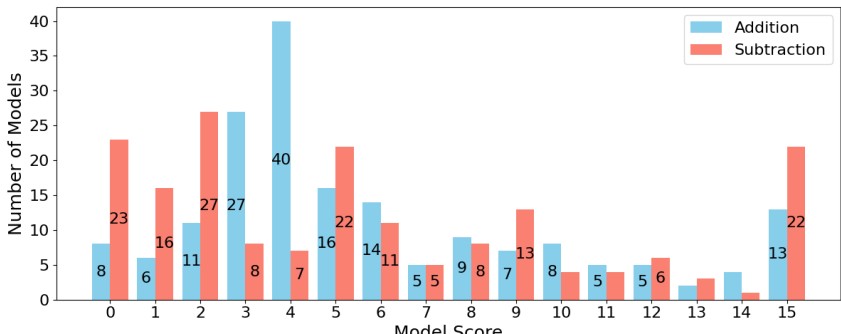

Figure 2: We score the addition and subtraction capability of 180 public LLMs. A score of 5 means the LLM handled two 5 digits numbers correctly but failed with 6 digit numbers. 7% of addition and 12% of subtraction models get the maximum test score 15. The top models can call external tools.

has ~10M parameters—over four orders of magnitude smaller than GPT-3—and trains to near-perfect accuracy in under an hour on a single GPU, providing an efficient, reproducible, and interpretable testbed for mechanistic analysis.

We introduce a novel, simple, mathematically exact algorithm for n-digit addition (Figure 1). The algorithm respects the transformer's left-to-right processing constraint, handling high-value digits before low-value ones. We also introduce a novel, exact algorithm for subtraction.

We then ask whether our trained models actually implement these algorithms. To do so, a model must realize each algorithmic subtask. We hypothesize the existence of specific calculation "nodes" capable of carrying out these subtasks, and develop code to automatically search for candidate nodes. We evaluate these candidates through systematic ablations. Each ablation is designed around a specific subtask, allowing us to precisely predict how the model's output should change when we intervene. The predictions match perfectly: intervening on the identified node produces exactly the expected effect. Every model we study contains nodes corresponding to all subtasks, though in some cases a single logical role is distributed across two attention heads.

The algorithms also impose strict constraints on node ordering and relationships. For example, certain nodes must appear earlier in the computation sequence, while others depend on them downstream (see Figure 1). We test these constraints and find they are consistently satisfied across seeds, model sizes, and digit ranges. Together, these results show that all our accurate models converge on the same algorithmic solution.

To make this analysis reproducible, we release a general-purpose interpretability library. `https://anonymous.4open.science/r/quanta_mech_interp-F9E5`. It provides tools for (1) node characterization, search, and ablation, (2) systematic testing of node relationships and constraints, and (3) visualizations. It is a foundational tool used by this project `https://anonymous.4open.science/r/quanta_maths-6413`

**Our contributions are fivefold**: (i) a large-scale survey of LLM arithmetic capabilities, (ii) enriched datasets for addition and subtraction, (iii) 49 small transformer models trained to near-perfect accuracy, (iv) human-comprehensible exact addition and subtraction algorithms and (v) a reusable interpretability toolkit demonstrating that these models implement the algorithms.

We conclude that transformers can implement exact n-digit addition and subtraction through interpretable circuits. With appropriate training and data, larger LLMs could in principle adopt the same mechanisms.

## 2 RELATED WORK

**Mechanistic understanding of arithmetic in transformers**. Early work by Nanda et al. (2023) revealed that one-layer models implement modular addition through discrete Fourier transforms, converting addition into rotations in frequency space - showing transformers can find unintuitive

algorithms for basic mathematics. Quirke & Barez (2024) detailed an algorithm for n-digit addition in a single-layer model that achieved 99% accuracy, identifying a cascading carry-one failure mode. Subsequent work on training dynamics (Musat, 2024) and the finding that even random transformers exhibit algorithmic capabilities (Zhong & Andreas, 2024) suggest architectural biases toward systematic computation.

**Architectural modifications and specialized models**. Several approaches achieve near-perfect arithmetic accuracy through architectural changes. Reformatting inputs to present least-significant digits first by adding position embeddings and zero-padding strategies (McLeish et al., 2024). Cho et al. (2024) align computational flow with carry propagation. Specialized models demonstrate superior performance: MathGLM's 10M parameter model achieves 100% accuracy on integer addition through step-by-step reasoning (Yang et al., 2023), while Qiu et al. (2024) reached 99.9% accuracy on 5-digit multiplication with a tiny transformer that outperforms GPT-4. Subtraction is much less studied, though Zhang et al. (2024) identified symmetries between addition and subtraction circuits, with the same attention heads handling both operations.

**Production systems versus specialized models**. The Claude 3.5 Haiku circuit analysis Lindsey et al. (2025) reveals how a frontier model implements addition. Rather than the clean n-digit algorithms found in specialized models, Claude employs a "bespoke" circuit combining lookup tables for memorized facts with magnitude estimation pathways. This distributed circuit achieves perfect accuracy on 2-digit addition using mechanisms distinct from both human methods and specialized models. The gap between Claude's complex implementation on short addition questions and the elegant n-digit algorithms in specialized models illustrates a fundamental challenge: production LLMs trained on diverse tasks develop hybrid strategies rather than the interpretable, generalizable algorithms that specialized models achieve.

## 3 METHODOLOGY

Transformer models learn addition algorithms different from traditional human methods. We define an alternative, mathematically-equivalent framework for addition and demonstrate our models implement this approach.

### 3.1 MATHEMATICAL FRAMEWORK

For addition and subtraction of two $n$-digit numbers, we use the Fig. 3 notation. (Detail in App. A).

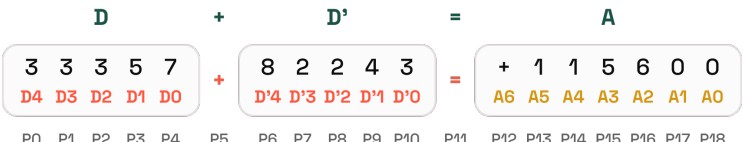

Figure 3: For 5-digit addition and subtraction, our notation for the main input tokens is $D4, ..., D0$ and $D'4, ..., D'0$. For output tokens it is $A6, .., A0$. For n-digits, we use the notation $D_n, .., D0, D'_n, .., D'0$ and $A_n, .., A0$.

### 3.2 ADDITION ALGORITHM DESCRIPTION

For addition, aligned with Quirke & Barez (2024), we define the "Base Add" subtask $SA_n$ to compute the digit-wise sum modulo 10:

$$SA_n = (D_n + D'_n) \mod 10 \tag{1}$$

To solve $n$-digit addition with high accuracy, the framework must handle carry bits that cascade through multiple digits. For an autoregressive model to correctly predict the first answer digit in "555+448=+1003", as "1", it must cascade the carry bit from the *rightmost* digit to the *leftmost* digit in a single forward pass. This is particularly challenging for an autoregressive model that processes tokens from left to right. Aligned to this challenge we define two new subtasks.

First, we introduce subtask ST. It classifies a digit pair sum as *definitely* causing a carry (e.g. 6+7), definitely *not* causing a carry (e.g. 2+3), or *possibly* causing a carry (e.g. 5+4). The 5+4 case is uncertain as the digits sum to 9 and a carry from the next-lower-value digit would cause a carry:

$$\underbrace{ST_n}_{(D_n, D'_n)} = \begin{cases} 1 & \text{if } (D_n + D'_n) \geq 10 \\ 0 & \text{if } (D_n + D'_n) \leq 8 \text{ or } n = 0 \\ U & \text{otherwise} \end{cases} \qquad \begin{array}{l} \text{Note ST0 is always 0 or 1.} \\ \text{Sums to 9, so carry is uncertain} \end{array} \qquad (2)$$

The "TriAdd" function combines two $ST_n$ style values X and Y. Only if *both* X and Y are U, does TriAdd return U (there is still uncertainty), otherwise it returns 0 or 1 (the carry bit value is known):

$$\underbrace{\text{TriAdd}}_{(X,Y)} = \begin{cases} Y & \text{if } X = U \\ X & \text{otherwise} \end{cases} \qquad (3)$$

Finally, we introduce the SV subtask that uses TriAdd to handle multidigit carry cascades. We describe $SV_n$ using the first 3 examples:

$$SV0 = ST0 \qquad (4)$$
$$SV1 = \text{TriAdd}(ST1, ST0) \qquad (5)$$
$$SV2 = \text{TriAdd}(\text{TriAdd}(ST2, ST1), ST0) \qquad (6)$$

The highlighted (solid-color) part of Figure 1 illustrates the SV2 calculation. Note that the calculation first combines ST2 and ST1 (the higher value digits) before combining the result with ST0. This formulation mirrors the left to right order in which the model processes tokens i.e. from higher value digits to lower value digits. The last term in all $SV_n$ calculations is ST0 which is always 0 or 1, so all $SV_n$ evaluate 0 or 1. That is, at the end of the $SV_n$ calculation any uncertainty has been resolved.

The $SV_n$ are perfectly accurate multidigit cascade carry values. Their calculation occurs across multiple tokens and is completed by the "=" token. Combining the $SV_n$ values with the $SA_n$ values as shown in Figure 1 gives perfectly accurate answer tokens.

### 3.3 SUBTRACTION ALGORITHM DESCRIPTION

Our subtraction algorithm mirrors our addition algorithm, replacing addition's "cascading carry one" with subtraction's "cascading borrow one". Both operations use the same iterative approach to reducing "cascading" uncertainty token by token.

We introduce "Base Diff" $MD_n$ defined as $D_n$- $D'_n$ modulo 10 (paralleling $SA_n$). We also introduce $MB_n$ for the single-digit "borrow one" case (paralleling $ST_n$):

$$\underbrace{MB_n}_{(D_n, D'_n)} = \begin{cases} 1 & \text{if } D_n < D'_n \text{ (borrow one)} \\ 0 & \text{if } D_n > D'_n \text{ (no borrow)} \\ U & \text{if } D_n = D'_n \text{ (uncertain)} \end{cases} \qquad (7)$$

Finally, we introduce $MV_n$ which parallels $SV_n$ to handle the "cascading borrow one" edge case. With this formulation, the addition and subtraction algorithms have the same structure (just replacing $SA_n$ with $MD_n$, $ST_n$ with $MB_N$, and $SV_n$ with $MV_N$).

Subtraction poses an additional algorithmic difficulty - some questions give a positive answer (e.g. 325-129=+196) and others a negative answer (e.g. 325-329=-004). Note that all the answer digits are different in the above examples. While there are multiple ways this could be handled mathematically, we introduce "Neg Diff" $ND_n$ defined as $D'_n$- $D_n$ modulo 10 (the opposite of $MD_n$). At the "=" token, the $MV_n$ avalues are known, and the algorithm selects $MD_n$ or $MD_n$ values as appropriate.

In Fig. 4, the value $MV2$ determines the answer sign. While the following calculations could use $MV2$ too, in fact they attend to the answer sign itself. We introduce the subtask SGN to reflect this.

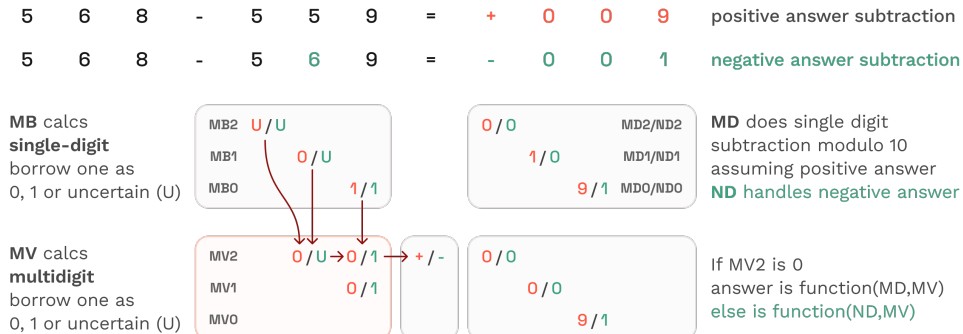

Figure 4: Our subtraction algorithm parallels our addition algorithm but it refines "cascading borrow one" uncertainty over multiple tokens. The refined $MV2$ value determines both the answer sign ("+" or "-") and which digit values ($MD_n$ or $ND_n$) to use for the final output.

### 3.4 MIXED ALGORITHM DESCRIPTION

For mixed models handling both operations, we add the subtask OPR meaning "attends to the prompt operation token". The mixed models uses OPR and SGN to determine whether to output addition, positive-subtraction, or negative-subtraction answer digits.

## 4 EXPERIMENTS

### 4.1 TRAINING MODELS

Our models' vocabulary consisted of 14 tokens: the digits 0-9 plus the mathematical operators +, -, =, *, and /. (Also see App. C). We created an infinite training dataset enriched with rare numerical combinations (mainly cascading use cases). (Also see App. D)

We trained addition-only, subtraction-only, and mixed (addition and subtraction) models. We systematically explored different model architectures and settled on 2 layers and 3 or 4 attention heads. (Also see App. F)

We trained used a batch size of 64, learning rate of 0.00008, and weight decay of 0.1. The loss function was defined as the mean negative log likelihood across all output tokens. Training was stopped when the loss was very low e.g. $2 \times 10^{-8}$. (Also see App. G, Fig. 5, Fig. 9).

Each model's accuracy was tested on 1 million questions. Robust statistical accuracy ranges were calculated, but for ease of presentation, we say a model has 99.999% accuracy if it got less than 10 of the 1 million questions wrong. Most models achieved this. The resulting 49 models, including detailed accuracy results, are summarized in Tab. 5 and Tab. 6.

#### 4.1.1 TRAINING MIXED MODELS

To explore how transformers can learn multiple arithmetic operations simultaneously, we conducted experiments with models capable of both addition and subtraction. We were particularly interested in whether knowledge from addition-specialized models could transfer to this expanded task.

We trained 14 mixed models with various architectures, including 6- and 10-digit models with 2-3 layers and 3-4 attention heads (see Tab.6). Each model was initialized using weights from a highly accurate addition-only model, specifically copying the weights into the first two layers and first three heads of the mixed model. During training, we used a curriculum of 80% subtraction and 20% addition examples, using enriched data for both operations (See Fig. 6).

This initialization approach proved effective, with most models quickly achieving high accuracy on both operations. We explored alternative training strategies, such as periodically "freezing" the inserted attention heads or MLPs by restoring their original addition-model weights every 100 training steps. However, these more constrained approaches resulted in lower accuracy, suggesting that some

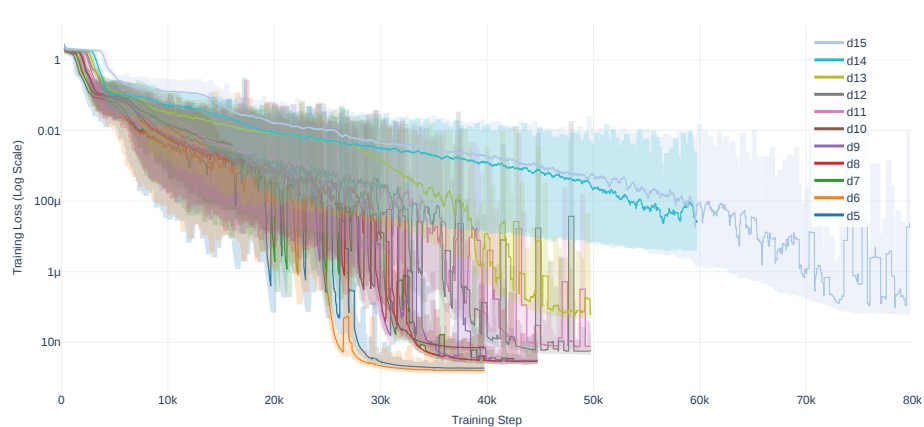

Figure 5: The 5- to 15-digit 2-layer 3-head addition models have very low loss. With more digits training takes longer. Details in Tab.5.

flexibility in modifying the initialized weights was necessary for optimal performance (Also see App. K)

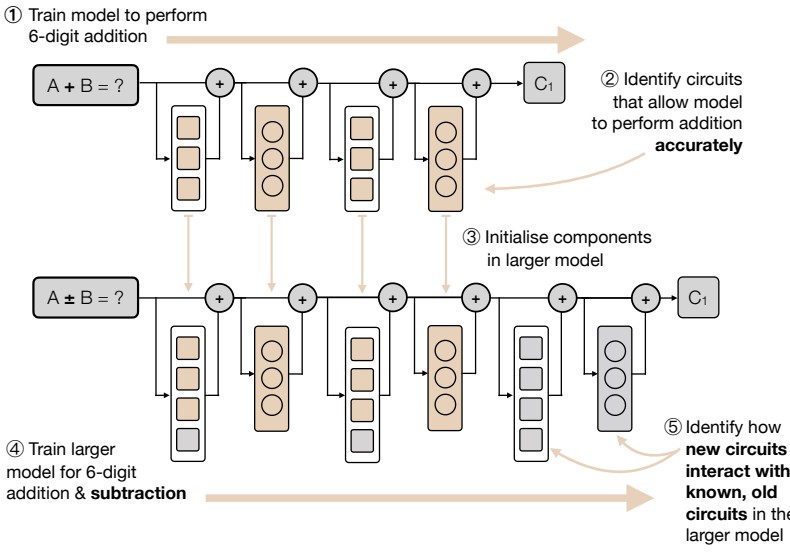

Figure 6: Mixed model initialization and analysis: (1) Train an accurate 5-digit addition model. (2) Reverse-engineer it to identify the implemented algorithm. (3) Insert its attention head and MLP weights (in brown) into a larger model. (4) Train the new model on 80% subtraction and 20% addition. (5) The resulting model predicts accurately and reuses the inserted addition circuits for both tasks.

We did a seed sensitivity analysis. App. L includes a visualization Fig. 15 of the range of training losses for different model categories.

## 4.2 EXPERIMENTAL RESULTS

To verify whether our models implement the algorithms described above, we developed a systematic framework for analyzing model internals. For each trained model, we investigated whether it contains the specific computational components our algorithms require and whether these components interact according to the algorithmic constraints.

### 4.2.1 Implementation Analysis Framework

Consider a 6-digit addition model. If it implements our algorithm, it must satisfy these constraints:

- **Functional accuracy**: The model must correctly add any two 6-digit numbers, selecting the unique correct answer from nearly 2 million possible answers.

- **Complete component coverage**: The model must implement the required subtasks for each digit position i.e. 6 $SA$, 6 $ST$, 6 $SV$, and 7 answer digit finalization nodes.

- **Component-level behavior**: Each node implementing a subtask must obey certain conditions. For instance, a node implementing $ST2$ must:
  - Attend primarily to tokens $D_2$ and $D'_2$ (post-softmax attention $> 0.01$)
  - Be positioned after $D'_2$ but before the "=" token
  - A Primary Component Analysis (PCA) should show 3 distinct clusters. (See Fig. 7.)
  - Given three test question sets, created specifically to trigger the 3 $ST2$ values, each question set should align with exactly one of the PCA clusters.

- **Algorithmic ordering constraints**: Node locations must respect computational dependencies. Since SV1 = TriAdd(ST1, ST0), a node computing SV1 must be positioned after nodes computing ST1 and ST0. Our 6-digit model has over 30 such ordering constraints.

- **Causal verification through intervention**: When we identify a candidate node for a specific subtask (e.g. ST2), we can construct paired test questions differing only in that subtask's value. By intervening on the candidate node and swapping its activations between the test pairs, we can predict the model's output from nearly 2 million possible answers.

For models with more than 6 digits, the constraints list grows further. The subtraction and mixed algorithms each have a more extensive constraints list. With 49 models to test, we developed an automated testing framework to:

- Systematically search for nodes over each subtask and digit position combination retaining only those that satisfy all the constraints in our algorithm

- Compare the list of nodes found with the algorithmic requirements (including intra-node relationships).

- Visualize the results.

### 4.2.2 Addition model results

We tested a single 1-layer addition model. Our framework found the $SA_n$ and $SC_n$ (single-digit carry calculated during answer tokens) nodes as described by Quirke & Barez (2024) that give the model its 99% accuracy.

Testing our fifteen 2-layer addition models, we find all models contain the subtasks required by our addition algorithm, and satisfy the algorithmic requirements. This consistent pattern suggests we have identified a robust and generalizable algorithm that naturally arises when transformer models are trained on addition in our setup. There is some variability between models:

- In a given model, $ST_N$ is not as "tidy" as shown in Fig. 1. Different models select different attention heads to calculate say ST2. Redundantly, some models have two nodes that both calculate say ST1. See Tab. 1.

- Some models combine 2 attention heads (with the same token position and layer) to calculate say SA2. Many models only use one attention head.

- For ST nodes, the PCA trigrams have difference appearances in each attention head (- but all have 3 clusters aligned to the ST values. See Fig. 7.

- All models have $ST$ nodes, but some models also have and use $SC$ nodes. The $SC$ nodes are redundant: the $ST$ values are equivalent and the $SV$ values are superior! See Tab. 1.

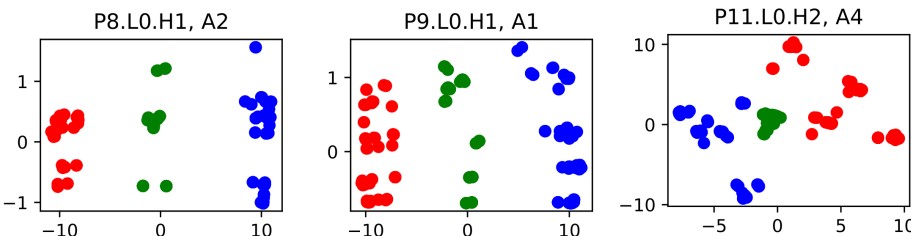

Figure 7: For a 5-digit addition model, these three sample PCAs each show 1 attention head at a token position tested for 1 answer digit. Each dot represents a question, colored by the question's expected ST value (0, 1, or U). While the PCA data varies across cases, the 3 ST classes form visually distinct clusters in each plot, suggesting that the attention head's activation patterns differ systematically based on ST value and that the model could potentially distinguish between these 3 cases.

Table 1: For a sample 5-digit addition model, we show a compacted location map for the $SA$, $SC$ and $ST$ subtasks. Interestingly 1) the $ST$ nodes are in a semi-random order 2) the second $ST1$ node is redundant 3) the model uses $SC$ nodes which are redundant 4) each $SA$ subtask is shared across two attention heads.

| | (P11) D'1 | (P12) D'0 | (P13) = | (P14) + | (P15) A6 | (P16) A5 | (P17) A4 | (P18) A3 | (P19) A2 | (P20) A1 |
|---|---|---|---|---|---|---|---|---|---|---|
| **L0H0** | ST2 | ST3 | ST1 | ST4 | SC4 | SC3 | SC2 | SC1 | SC0 | |
| **L0H1** | ST1 | | ST0 | | SA5 | SA4 | SA3 | SA2 | SA1 | SA0 |
| **L0H2** | | | | ST5 | | | | | | |

### 4.2.3 MIXED MODEL RESULTS

Testing our accurate "mixed" (both addition and subtraction) models, we find all models contain the subtasks required by our addition algorithm, and satisfy the algorithmic requirements.

The mixed models have the same types of variability that the addition models have.

**Evolution of Polysemantic Nodes.** Zhang et al. (2024) identified symmetries between addition and subtraction circuits, with the same attention heads handling both operations. We expand on this finding.

Consider the mixed models that we initialized with accurate, smaller addition models before training. The majority of inherited nodes became polysemantic during training. Rather than developing entirely new circuits for subtraction, the models adapted their addition circuits to handle both operations. For example, a node that originally only performed Base Add ($SA$) often learns to handle three subtasks: addition ($SA$), positive-answer subtraction diff ($MD$), and negative-answer subtraction diff ($ND$). (Refer to Tab. 2.)

We believe that this adaptation was likely because these three operations share a common structure: each maps a pair of digits (10 possibilities for $D_n \times$ 10 possibilities for $D'_n$ = 100 input cases) to a single-digit result (10 possible outputs). Similarly, some models adapted nodes that originally handled carry bits ($ST$) to also process borrow bit operations ($MT$).

## 5 CONCLUSION

Our analysis of 49 transformer models demonstrate their ability to learn highly accurate mathematical algorithms, achieving over 99.999% accuracy on addition tasks up to 15 digits. By investigating how models implement computations using attention heads and MLPs, we uncovered novel computational approaches distinct from traditional human calculation methods. Parameter transfer experiments revealed that features originally specialized for addition could be repurposed for subtraction, with many nodes becoming polysemantic.

Table 2: For this mixed model, in the last 5 tokens, polysemantic attention heads simultaneously generate outputs for the three question classes (S, M and N). Other heads calculate the question class by attending to the question operation (OPR) token and the answer sign (SGN) token. We assume the MLP layers then select the output appropriate for the class.

| | (P9)
D'3 | (P10)
D'2 | (P11)
D'1 | (P12)
D'0 | (P13)
= | (P14)
A7 | (P15)
A6 | (P16)
A5 | (P17)
A4 | (P18)
A3 | (P19)
A2 | (P20)
A1 |
|---|---|---|---|---|---|---|---|---|---|---|---|---|
| **L0H0** | MT4 | | | MT3 | | ST4
MT4
OPR | SC4 | SC3 | SC2
NB2 | SC1
MB1
NB1 | SC0
MB0
NB0 | OPR
SGN |
| **L0H1** | ST4 | ST2
MT2 | ST1
MT1 | ST3 | ST0
MT0 | | SA5
MD5 | SA4 | SA3 | SA2 | SA1 | SA0 |
| **L0H2** | | | | | | ST5
OPR
SGN | SA4
ND5 | MD4
ND4 | MD3
ND3 | MD2
ND2 | MD1
ND1 | MD0
ND0 |
| **L0H3** | | | | | | | OPR
SGN | OPR
SGN | OPR
SGN | OPR
SGN | OPR
SGN | OPR
SGN |

Table 3: Mixed models re-use most inserted addition-model nodes. Many inserted nodes become polysemantic during training - performing addition, positive-answer subtraction **and** negative-answer subtraction subtasks simultaneously. For a sample mixed model that uses 96 nodes and had 48 nodes inserted, this table shows inserted node reuse.

| | Used | | Inserted | |
|---|---|---|---|---|
| **Question class** | **#** | **%** | **#** | **%** |
| All questions | 96 | | 48 | |
| Addition | 61 | 64% | 42 | 88% |
| Positive-answer subtraction | 70 | 73% | 40 | 83% |
| Negative-answer subtraction | 53 | 55% | 29 | 60% |

Our research provides insights into how transformers learn precise algorithmic computations across related tasks, developing a reusable library of mechanistic interpretability tools that advances our understanding of neural network computational learning.

# 6 LIMITATIONS AND FUTURE WORK

While we identify and test the functional *role* of each component in the mixed model algorithm, we do not analyze the specific *data representations* and *transformations* used by polysemantic nodes, SGN nodes, and OPR nodes in the residual stream, nor the detailed mechanisms by which MLP layers process this information. Although our algorithmic-level analysis may be more generalizable across models, this approach leaves gaps in understanding how the identified components actually encode and manipulate information.

Our automated framework for discovering algorithm subtasks in models, while instrumental in accelerating our research, has limitations. Some aspects are specific to our math models. Some aspects are generic and have already been used in the published paper by Harrasse et al. (2025).

Fully reverse engineering neural networks faces significant challenges, including distributed parallel computations difficult to trace to specific components. We made progress on an alternative approach: a declarative language describing algorithms in terms of subtasks and a framework to test these descriptions against models. However, more work is needed to refine this method and quantity our certainty when comparing evidence gathered from models against an algorithmic description.

Our models may be useful to researchers investigating the universality of representations / circuits across models.

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

## A  APPENDIX: TERMINOLOGY AND ABBREVIATIONS

These terms and abbreviations are used in this paper and the associated Colabs and python code:

- **Pn** : Model (input or output) token position. Zero-based. e.g. **P**18, **P**18L1H0
- **Ln** : Model layer n. Zero-based. e.g. P18**L**1H2
- **Hn** : Attention head n. Zero-based. e.g. P18L1**H**2
- **Mn** : MLP neuron n. Zero-based
- **PnLnHn** : Location / name of a single attention head, at a specified layer, at a specific token position

- **PnLnMn** : Location / name of a single MLP neuron, at a specified layer, at a specific token position

- **D** : First number of the pair question numbers

- **Dn** : nth numeric token in the first question number. Zero-based. D0 is the units value

- **D'** : Second number of the pair question numbers

- **D'n** : nth token in the second question number. Zero-based. D0 is the units value

- **A** : Answer to the question (including answer sign)

- **An** : nth token in the answer. Zero-based. A0 is the units value. The highest token is the "+" or "-" answer sign

- **S** : Prefix for Addition. Think S for Sum.

- **SA** : Base Add. An addition subtask. $SA_n$ is defined as (Dn + D'n) % 10. e.g. 5 + 7 gives 2

- **SC** : Carry One. An addition subtask. $SC_n$ is defined as Dn + D'n >= 10. e.g. 5 + 7 gives True

- **ST** : TriCase. An addition subtask. Refer paper body for details

- **M** : Prefix for Subtraction with a positive answer. Think M for Minus. Aka SUB

- **MD**: Basic Difference. A subtraction subtask. $MD_n$ is defined as (Dn - D'n) % 10. e.g. 3 - 7 gives 6

- **MB**: Borrow One. A positive-answer subtraction subtask. $MB_n$ is defined as Dn - D'n < 0. e.g. 5 - 7 gives True

- **N** : Prefix for Subtraction with a negative answer. Think N for Negative. Aka NEG

- **ND** : Basic Difference. A negative-answer subtraction subtask. $ND_n$ is defined as (Dn - D'n) % 10. e.g. 3 - 7 gives 6

- **NB** : Borrow One. A negative-answer subtraction subtask. $NB_n$ is defined as Dn - D'n < 0. e.g. 5 - 7 gives True

- **OPR** : Operator. A subtask that attends to the + or - token in the question (which determines whether the question is addition or subtraction).

- **SGN** : Sign. A subtask that attends to the first answer token, which is + or -

- **PCA** : Principal Component Analysis

- **EVR** : Explained Variance Ratio. In PCA, EVR represents the percentage of variance explained by each of the selected components.

## B   APPENDIX: ETHICS STATEMENT

Our work aims to explain the inner workings of transformer-based language models, which may have broad implications for a wide range of applications. A deeper understanding of generative AI has dual usage. While the potential for misuse exists, we discourage it. The knowledge gained can be harnessed to safeguard systems, ensuring they operate as intended. It is our sincere hope that this research will be directed towards the greater good, enriching our society and preventing detrimental effects. We encourage responsible use of AI, aligning with ethical guidelines.

## C   APPENDIX: VOCABULARY

Each digit is represented as a separate token. (Liu & Low, 2023) state that LLaMa's "remarkable arithmetic ability ... is mainly atributed to LLaMA's consistent tokenization of numbers". The model's vocabulary contains 14 tokens ( 0, .., 9, +, -, =, *, / ) to enable this and planned future investigations.

## D    APPENDIX: TRAINING DATA

Training uses a new batch of data each step (aka Infinite Training Data) to minimize memorization. Depending on the configuration, each training run processes 1 to 4 million training datums. For the 5-digit addition problem there are 100,000 squared (that is 10 billion) possible questions. So the training data is much less than 1% of the possible problems.

Addition and subtraction include rare edge cases.   For example, these cascades (e.g. 55555+44446=100001, 54321+45679=1000000, 44450+55550=10000, 1234+8769=10003) are exceedingly rare. The data generator was enhanced to increase the frequency of all known edges cases. This leads to lower model loss.

We enriched 60% of training data based on these edge cases (leaving the other 40% of training data random): (1) The initial model failed at cascading carry ones in addition. To rectify this, we randomly selected one operand and modified a random subset of digits in that operand to make the selected digit-position sum to 9, increasing the likelihood of a cascading carry one. (2) The initial model was worse at the subtraction task when the answer was negative, so we added 1 to each second operand digit (that was 8 or less) to increase the frequency of negative answers. (3) When the operands were identical, the initial model predicted -000000 instead of +000000. We increased the frequency of this case. For example, for 6 digit questions, we increased the frequency from 0.0001% to 0.6%.

## E    APPENDIX: SURVEYING LLM ADDITION CAPABILITY

LLM Gateways provide access to hundreds of LLMs.   We used the Martian Gateway ( https://app.withmartian.com/ ) to quickly and cheaply access 193 LLMs ( https://withmartian.github.io/llm-adapters/ ) ranging in size from 1 to 405 billion parameters. We tested their ability to perform addition accurately on the test questions containing cascading carry ones shown in Tab. 4. A model's score represents the longest question the model succeeded on before it failed a question. The results are shown in Fig. 2. Fig. 8 shows the size of the models seems uncorrelated to its ability to perform these questions.

Table 4: Addition prompts used to test LLMs' ability to handle the cascading carry one use case.

| Test Prompt | Correct Answer |
|---|---|
| Answer concisely: 6+5= | 11 |
| Answer concisely: 19+87= | 106 |
| Answer concisely: 774+229= | 1003 |
| Answer concisely: 6587+3416= | 10003 |
| Answer concisely: 22605+77398= | 100003 |
| Answer concisely: 532847+467159= | 1000006 |
| Answer concisely: 5613709+4386294= | 10000003 |
| Answer concisely: 72582383+27417619= | 100000002 |
| Answer concisely: 206727644+793272359= | 1000000003 |
| Answer concisely: 7580116456+2419883549= | 10000000005 |
| Answer concisely: 52449010267+47550989737= | 100000000004 |
| Answer concisely: 888522030597+111477969406= | 1000000000003 |

At time of writing, 4.76% of models scored 10 or more (specifically nousresearch/hermes-3-llama-3.1-405b, amazon/nova-pro-v1, anthropic/claude-opus-4-1-20250805, mistralai/magistral-medium-2506, openai/gpt-4o-mini-search-preview, openai/gpt-4o-search-preview, openai/o4-mini, qwen/qwen3-coder and tencent/hunyuan-a13b-instruct). Some of these models explicitly acknowledge the use of external tools. For others it is not obvious whether they use tools. The three Chat GPT 5 models tested scored 4, 4 and 7. After publication, we will make public the Jupyter notebook used to perform the test. It takes less than 20 minutes and costs less than 1 USD to run.

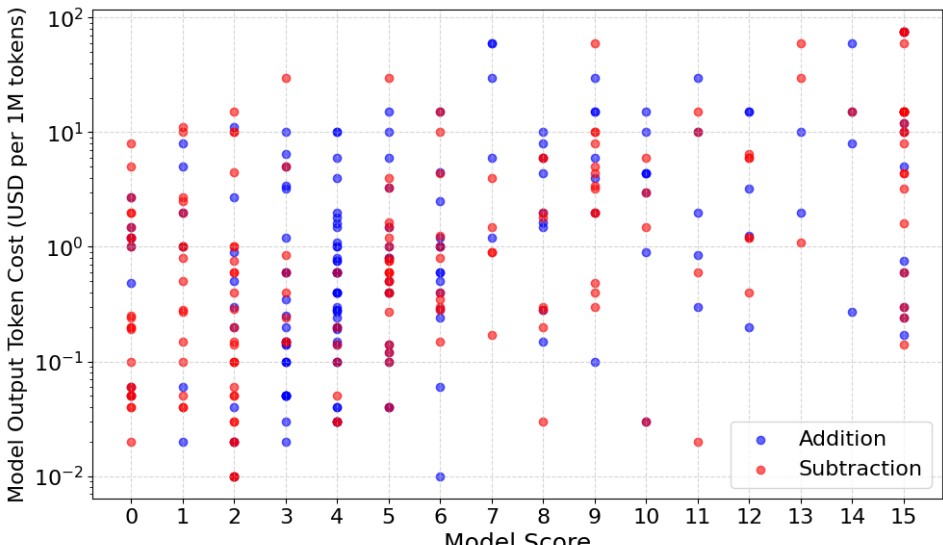

Figure 8: This scatter plot shows each model's addition and subtraction score against the model output token cost (an approximate metric for the model size). The maximum possible score is 15.

## F    APPENDIX: TRAINING SETUP

Addition, subtraction and mixed (addition and subtraction) training experiments were done in a Colab notebook. The Colab runs on a T4 GPU. Each training run takes up to 60 mins. The key parameters (and their common configurations) are:

- n_layers = 1, 2 or 3: Number of layers.

- n_heads = 3 or 4: Number of attention heads.

- n_digits = 5 .. 15: Number of digits in the question.

The Colabs will be made available on publication.

While we wanted a very low loss models, we also wanted to keep the model compact - intuiting that a smaller model would be easier to understand than a large model. Here are the things we tried to reduce loss that **didn't** work:

- Increasing the frequency of hard (cascading carry one) examples in the training data so the model has more hard examples to learn from. This improved training speed but did not reduce loss.

- Increasing the number of attention heads from 3 to 4 or 5 (while still using 1 layer) to provide more computing power.

- Changing the question format from "12345+22222=" to "12345+22222equals" giving the model more prediction steps after the question is revealed before it needs to state the first answer digit.

- Inserting "+" in the answer format (e.g. "12345+22222=034567" becomes "12345+22222=+034567" had no impact on accuracy or the algorithm.

- With n_layers = 1 increasing the number of attention heads from 3 to 4.

- Changing the n_layers to 2 and n_heads to 2.

The smallest model shape that did reduce loss significantly was 2 layers with 3 attention heads.

## G   APPENDIX: MODEL LOSS

The model defaults to batch size = 64. The loss function is simple:

- Per Digit Loss: For "per digit" graphs and analysis, for a given answer digit, the loss used is negative log likelihood.

- All Digits Loss: For "all answer digits" graphs and analysis, the loss used is the mean of the "per digit" loss across all the answer digits.

In our experimental models, the number of digits in the question varies from 5 to 15, the number of layers varies from 1 to 4, the number of heads varies from 3 to 4. Each experimental model's loss is detailed in Tabs. 5 and 6.

During training, the models use an AdamW optimizer with a learning rate (LR) of 0.00008, weight decay of 0.1 and betas of (0.9, 0.98). During the first 5th of training the LR is linearly increased from 0.01 * LR to LR to help stabilize training. During the remainder of training, cosine annealing is used.

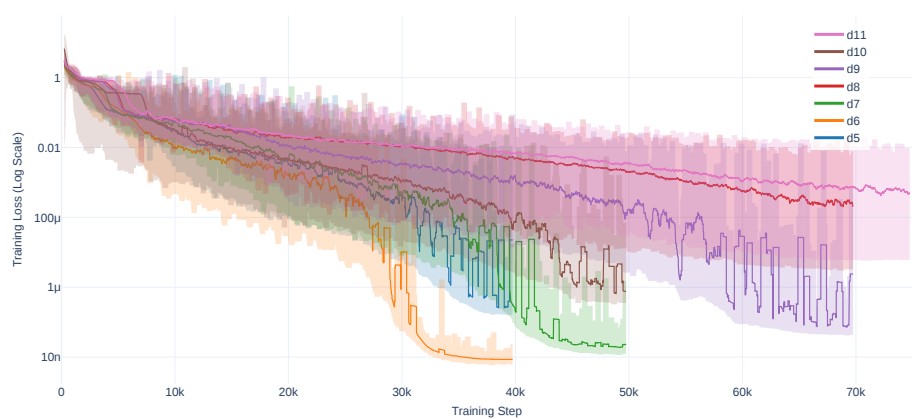

Figure 9: The 5- to 8-digit initialized mixed models have very low loss and >99.999% accuracy. With more digits the model loss and accuracy is worse. Details in Tab.6.

## H   APPENDIX: EXPERIMENTAL MODELS

Forty-nine models were trained and analyzed (Refer Tabs. 5 and 6). The models and analysis output will be made available on HuggingFace on publication to support further research in AI Safety.

For each model the 'QuantaMathsTrain' Colab notebook generates two files:

- A "model.pth" file containing the model weights

- A "training_loss.json" file containing configuration information and training loss data

While, for each model the 'QuantaMathsAnalysis' Colab notebook generates two more files:

- A "behaviors.json" file containing generic "behavior" facts learnt about the model by the Colab e.g. P18L0H0 attends to tokens D3 and D'3

- A "features.json" file containing maths-specific "feature" facts learnt about the model by the Colab e.g. P18L0H0 performs the SC3 subtask.

Table 5: Addition-only models studied. The number of addition failures per million questions is shown. The number of useful attention heads at token position and useful MLP layers at token position are shown - summarizing the data shown in figures like Fig 11.

| Digits | Layers | Heads | Train Steps | Train Seed | Train loss | Add Fails / M | Heads used | MLPs used | Clopper-Pearson 95% Interval |
|--------|--------|-------|-------------|------------|------------|---------------|------------|-----------|------------------------------|
| 5 | 1 | 3 | 30K | 372001 | 9.4e-2 | 12621 | 15 | 6 | [1.24e-2, 1.28e-2] |
| 5 | 2 | 3 | 15K | 372001 | 1.6e-8 | 0 | 30 | 16 | [0.00e+0, 3.69e-6] |
| 5 | 2 | 3 | 40K | 372001 | 2.0e-9 | 0 | 22 | 15 | [0.00e+0, 3.69e-6] |
| 6 | 2 | 3 | 15K | 372001 | 1.7e-8 | 2 | 31 | 17 | [2.42e-7, 7.22e-6] |
| 6 | 2 | 3 | 20K | 173289 | 1.5e-8 | 0 | 28 | 17 | [0.00e+0, 3.69e-6] |
| 6 | 2 | 3 | 20K | 572091 | 7.0e-9 | 0 | 35 | 17 | [0.00e+0, 3.69e-6] |
| 6 | 2 | 3 | 40K | 372001 | 2.0e-9 | 0 | 29 | 17 | [0.00e+0, 3.69e-6] |
| 7 | 2 | 3 | 45K | 173289 | 3.0e-9 | 0 | 31 | 20 | [0.00e+0, 3.69e-6] |
| 8 | 2 | 3 | 45K | 173289 | 3.0e-9 | 0 | 35 | 22 | [0.00e+0, 3.69e-6] |
| 9 | 2 | 3 | 45K | 173289 | 3.0e-9 | 0 | 54 | 27 | [0.00e+0, 3.69e-6] |
| 10 | 2 | 3 | 40K | 572091 | 7.0e-9 | 0 | 44 | 28 | [0.00e+0, 3.69e-6] |
| 11 | 2 | 3 | 50K | 173289 | 8.0e-9 | 2 | 56 | 29 | [2.42e-7, 7.22e-6] |
| 12 | 2 | 3 | 50K | 173289 | 5.0e-9 | 3 | 50 | 33 | [4.25e-7, 7.88e-6] |
| 13 | 2 | 3 | 50K | 173289 | 6.3e-8 | 1 | 66 | 31 | [2.53e-8, 5.57e-6] |
| 14 | 2 | 3 | 60K | 173289 | 5.5e-6 | 199 | 68 | 35 | [1.74e-4, 2.00e-4] |
| 15 | 2 | 3 | 80K | 572091 | 8.6e-8 | 10 | 93 | 58 | [4.71e-6, 1.67e-5] |

# I APPENDIX: COMPLEXITY

To analyze question difficulty, we categorized addition questions by the complexity of the computation required to solve the question, as shown in Tab. 8. The categories are arranged according to the number of digits that a carry bit has to cascade through.

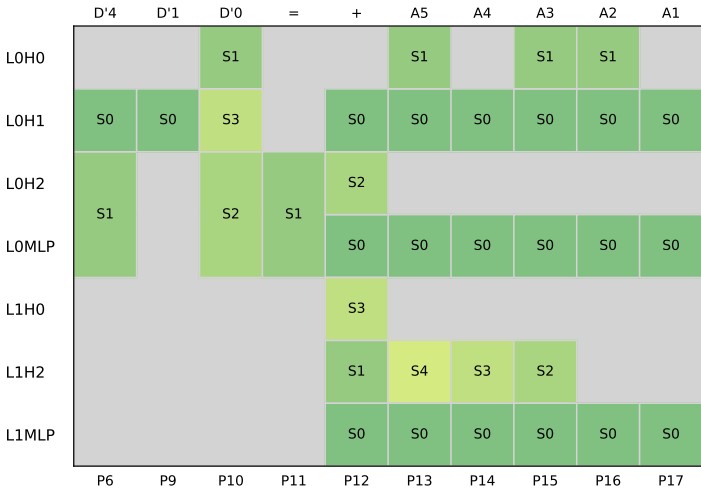

Figure 10: For a sample **5-digit** 2-layer 3-head **addition** model, this map shows a compacted view of all useful token positions (horizontally) and all useful attention heads and MLP layers (vertically) used in predictions as green cells. Each cell shows the simplest (lowest **complexity**) quanta S0, S1, etc impacted when we ablate each node. To answer S0 questions, only the S0 nodes are used. To answer S1 questions, S0 and S1 nodes are used, etc. The model only uses nodes in ten token positions.

Table 6: Subtraction-only and mixed models studied. The number of addition and subtraction failures per million questions is shown. The number of useful attention heads at token position and useful MLP layers at token position are shown - summarizing the data shown in figures like Fig 11.

| Digits | Layers | Heads | Train Steps | Train Seed | Train loss | Add Fails / M | Sub Fails / M | Heads used | MLPs used | Clopper-Pearson 95% Interval |
|---|---|---|---|---|---|---|---|---|---|---|
| \multicolumn{11}{c}{**Subtraction models**} |
| 5 | 2 | 3 | 30K | 372001 | 1.0e-3 | N/A | 3689 | 57 | 20 | [3.57e-3, 3.81e-3] |
| 6 | 2 | 3 | 30K | 372001 | 5.8e-6 | N/A | 2 | 37 | 21 | [2.42e-7, 7.22e-6] |
| 6 | 2 | 3 | 30K | 572091 | 5.8e-4 | N/A | 3889 | 58 | 21 | [3.77e-3, 4.01e-3] |
| 8 | 2 | 3 | 50K | 173289 | 4.0e-9 | N/A | 1 | 65 | 26 | [2.53e-8, 5.57e-6] |
| 8 | 2 | 3 | 50K | 371793 | 2.5e-5 | N/A | 487 | 71 | 28 | [4.45e-4, 5.32e-4] |
| 10 | 2 | 3 | 75K | 173289 | 2.0e-3 | N/A | 6672 | 101 | 37 | [6.50e-3, 6.72e-3] |
| 12 | 2 | 3 | 75K | 371793 | 3.4e-4 | N/A | 2175 | 96 | 32 | [2.08e-3, 2.25e-3] |
| \multicolumn{11}{c}{**Mixed models**} |
| 5 | 3 | 4 | 40K | 372001 | 9.0e-9 | 0 | 0 | 45 | 21 | [0.00e+0, 3.69e-6] |
| 6 | 3 | 4 | 40K | 372001 | 5.0e-9 | 1 | 0 | 54 | 26 | [2.53e-8, 5.57e-6] |
| 7 | 3 | 4 | 50K | 372001 | 2.0e-8 | 2 | 6 | 108 | 40 | [3.45e-6, 1.58e-5] |
| 8 | 3 | 4 | 60K | 173289 | 4.7e-8 | 0 | 7 | 123 | 45 | [2.81e-6, 1.44e-5 |
| 9 | 3 | 4 | 60K | 173289 | 3.2e-7 | 1 | 33 | 140 | 46 | [2.35e-5, 4.75e-5] |
| 10 | 3 | 4 | 75K | 173289 | 1.1e-6 | 2 | 295 | 143 | 53 | [2.65e-4, 3.53e-4] |
| 11 | 3 | 4 | 80K | 572091 | 3.9e-8 | 0 | 13 | 138 | 50 | [6.52e-6, 2.33e-5] |
| 12 | 3 | 4 | 85K | 572091 | 1.7e-8 | 2 | 10 | 167 | 55 | [8.68e-6, 2.72e-5] |
| 13 | 3 | 4 | 85K | 572091 | 9.5e-6 | 399 | 4164 | 197 | 64 | [4.41e-3, 4.70e-3] |
| \multicolumn{11}{c}{**Mixed models initialized with addition model**} |
| 5 | 2 | 3 | 40K | 572091 | 2.5e-7 | 1 | 52 | 49 | 20 | [5.35e-5, 8.48e-5] |
| 6 | 2 | 3 | 40K | 572091 | 2.4e-8 | 0 | 5 | 57 | 21 | [1.63e-6, 1.10e-5] |
| 6 | 3 | 3 | 40K | 572091 | 1.8e-8 | 0 | 3 | 70 | 35 | [4.25e-7, 7.88e-6] |
| 6 | 3 | 3 | 80K | 572091 | 1.6e-8 | 0 | 3 | 75 | 35 | [4.25e-7, 7.88e-6] |
| 6 | 3 | 4 | 40K | 372001 | 8.0e-9 | 0 | 0 | 72 | 26 | [0.00e+0, 3.69e-6] |
| 6 | 3 | 4 | 40K | 173289 | 1.4e-8 | 3 | 2 | 60 | 29 | [2.16e-6, 1.18e-5] |
| 6 | 3 | 4 | 50K | 572091 | 2.9e-8 | 0 | 4 | 79 | 29 | [8.11e-7, 9.28e-6] |
| 7 | 3 | 4 | 50K | 572091 | 1.8e-8 | 4 | 1 | 104 | 38 | [1.03e-6, 1.08e-5] |
| 8 | 3 | 4 | 70K | 572091 | 4.3e-5 | 50 | 1196 | 116 | 42 | [1.11e-3, 1.23e-3] |
| 9 | 3 | 4 | 70K | 572091 | 5.4e-8 | 1 | 4 | 160 | 50 | [8.11e-7, 9.28e-6] |
| 10 | 3 | 3 | 50K | 572091 | 6.3e-7 | 6 | 7 | 90 | 45 | [5.64e-6, 2.15e-5] |
| 11 | 3 | 4 | 75K | 572091 | 5.9e-5 | 11066 | 1120 | 141 | 47 | [1.19e-2, 1.22e-2] |
| \multicolumn{11}{c}{**Mixed models initialized with add model. Reset useful heads every 100 steps**} |
| 6 | 4 | 4 | 40K | 372001 | 1.7e-8 | 3 | 8 | 51 | 30 | [5.64e-6, 2.15e-5] |
| \multicolumn{11}{c}{**Mixed models inited with add model. Reset useful hds & MLPs every 100 steps**} |
| 6 | 4 | 3 | 40K | 372001 | 3.0e-4 | 17 | 3120 | 115 | 53 | [3.06e-3, 3.30e-3] |

## J  APPENDIX: N-DIGIT ADDITION

The addition models perform addition accurately. Visualizations that provided insights into the behavior of the model, aiding our interpretation of the algorithm, are below:

Some notes about the models:

- The models selected different attention heads in the early positions to use to do the same logical calculations.

- Some models use 2 attention heads per digit to do the $SA$ calculation, whereas some models only uses one (and so are more compact).

- The PCA trigrams have difference appearances in different models (but the same interpretable clusters). Refer Figures 7

Table 7: For a sample model, all nodes used in predictions are shown by token position (horizontally) and model layer (vertically), detailing the **answer digits** they impact. Here, the attention heads in token position P10 labelled A5..3 help predict the answer digits A3, A4 and A5. For all addition and mixed models studied, before the "=" token, each node often calculates data used to predict **multiple** answer digits. After the "=" token, all nodes in a given token position are used to predict a **single** answer digit.

| | (P6) D'4 | (P9) D'1 | (P10) D'0 | (P11) = | (P12) + | (P13) A5 | (P14) A4 | (P15) A3 | (P16) A2 | (P17) A1 |
|---|---|---|---|---|---|---|---|---|---|---|
| **L0H0** | | | | | | A4 | | A2 | A1 | |
| **L0H1** | | A5 | A5..3 | | | | A3 | | | A0 |
| **L0H2** | A5 | | | A5..1 | | | | | | |
| **L0MLP** | | | A5..2 | | A5 | A4 | A3 | A2 | A1 | A0 |
| **L1H0** | | | | | | | | | | |
| **L1H2** | | | | | | A4 | A3 | A2 | | |
| **L1MLP** | | | | | | | | | A1 | A0 |

Table 8: We categorise addition questions into non-overlapping "calculation complexity" quanta, ordered by increased computational difficulty (and decreasing occurrence frequency). Five-digit addition questions quanta are *S0* to *S5*. Ten-digit addition question quanta are *S0* to *S10*. *S10*'s frequency is $\sim 3e-4$ showing the need to enrich training data for rare edge cases.

| Name | Contains | Example | Freq |
|---|---|---|---|
| *S0* | *SA* | 11111+12345=23456 | $\sim$5% |
| *S1* | SA,SC | 11111+9=22230 | $\sim$21% |
| *S2* | SA,SCx2 | 11111+89=22300 | $\sim$34% |
| *S3* | SA,SCx3 | 11111+889=23000 | $\sim$28% |
| *S4* | SA,SCx4 | 11111+8889=30000 | $\sim$11% |
| *S5* | SA,SCx5 | 11111+88889=100000 | $\sim$2% |

- Per answer digit, some models use the $SC$ calculation, whereas some models optimize it out and rely solely on the $ST$ value (and so are more compact).

## K    APPENDIX: MIXED MODEL INITIALIZATION

We experimented with three approaches to re-using the trained addition model in the "mixed" (addition and subtraction) model:

- Initialize Only: Initialize the untrained mixed model with the addition model weights before training begins.

- Freeze Attention: As per "Initialize Only", but also every 100 training steps recopy the attention head weights from the addition model into the partially-trained mixed model.

- Freeze All: As per "Initialize Only", but also every 100 training steps recopy the entire addition model (attention heads and MLP layers) into the partially-trained mixed model.

Our intuition was that "Initialize Only" would give the mixed model the most freedom to learn new algorithms, but that the "Freeze Attention" and "Freeze All" approaches would make the resulting trained mixed model easier to interpret (as we could reuse our addition model insights).

After experimentation we found that the "Initialize Only" approach was the only one that quickly trained to be able to do both addition and subtraction accurately. We concluded that the other two methods constrain the model's ability to learn new algorithms too much.

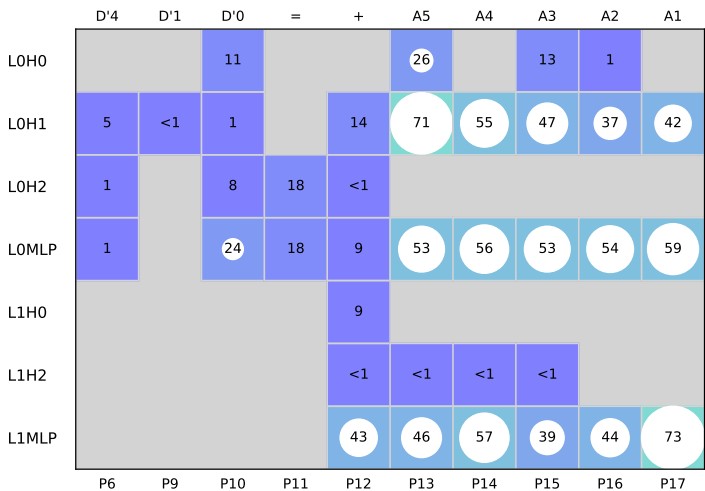

Figure 11: This map shows the **% of enriched questions** that fail when we ablate each node in a **5-digit** 2-layer 3-head addition model. The model only uses nodes in token positions P6 to P17 (i.e. tokens D'4 to A1). Lower percentages correspond to rarer edge cases. The grey space represents nodes that are not used by the model.

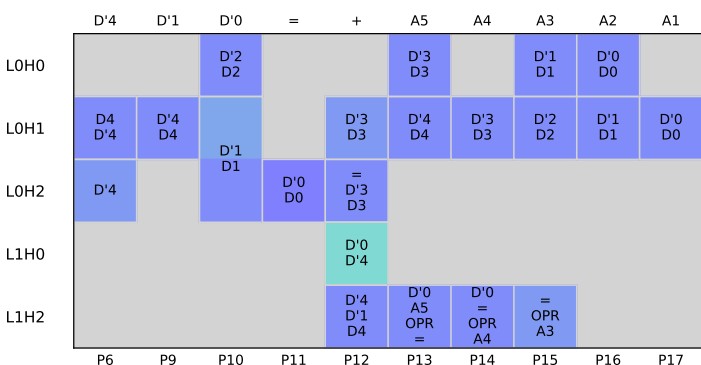

Figure 12: This map shows the input tokens each attention head attends to at each token position in a **5-digit** 2-layer 3-head addition model. At token position P12 the model predicts the first answer digit A5. **All** digit pairs (e.g. D2 D'2) are attended to by P12.

We also experimented with "where" in the model we inserted the addition (6-digit, 2-layer, 3-head) model into the slightly larger (6-digit, 3-layer, 4-head) mixed model. That is, do we initialize the first 2 layers or the last 2 layers of the mixed model? Also do we initialize the first 3 attention heads or the last 3 attention heads of the mixed model? Our intuition was that initializing the first layers and heads would be more likely to cause the model to re-use the addition circuits adding interpretability, so we used this approach.

## L   APPENDIX: SEED SENSITIVITY

An analysis of the sensitivity of 48 models to the initial seed was performed (This analysis excluded one model - the inaccurate 1-layer addition model that reproduces the Quirke & Barez (2024) paper results). Fig. 15 shows the results. We conclude:

- The Addition models are the most stable - that is they are not sensitive to the seed value.

- The other categories (Subtraction, Mixed, Mixed+Init, and Mixed+Reset) show relatively low to moderate sensitivity.

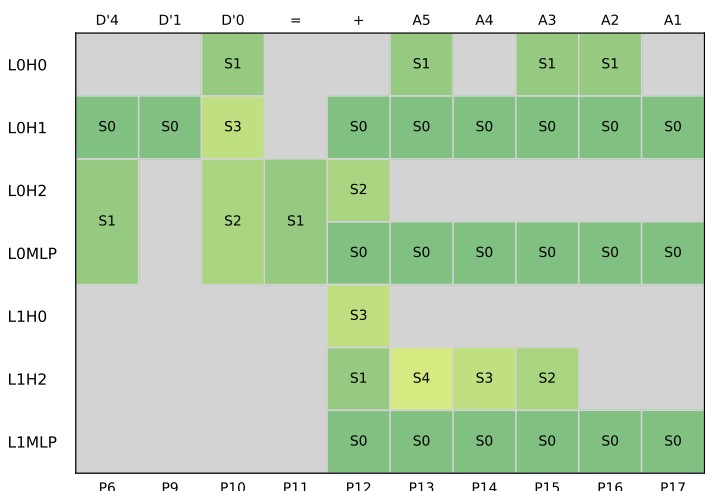

Figure 13: This map shows the simpliest (lowest **complexity**) quanta S0, S1, etc impacted when we ablate each node in the **5-digit** 2-layer 3-head **addition** model. To answer S0 questions, only the S0 nodes are used. To answer S1 questions, S0 and S1 nodes are used, etc.

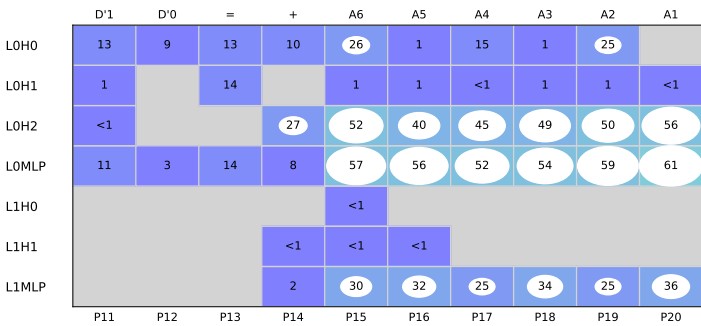

Figure 14: This map shows the **% of questions** that fail when we ablate each node in the **6-digit** 2-layer 3-head addition model. The model only uses nodes in token positions P11 to P20. Lower percentages correspond to rarer edge cases. The grey space represents nodes that are not useful.

- The higher average loss for Subtraction models show that the models find it harder to learn Subtraction is isolation.
- The higher average loss for Mixed+Reset models show that the this type of intervention during training makes it harder for the models to learn

## M APPENDIX: N-DIGIT SUBTRACTION

The mixed models perform addition and subtraction accurately. Visualizations that provided insights into the behavior of the model, aiding our interpretation of the algorithm, are below:

Some notes about the mixed models:

- All the notes about the addition model (above) also apply to the mixed model.
- The model contains a new subtask that stands out: The algorithm relies on calculations done at token position P0, when the model has only seen one question token! What information can the model gather from just the first token? Intuitively, if the first token is a "8" or "9" then the first answer token is more likely to be a "+" (and not a "-"). The model uses this heuristic even though this probabilistic information is sometimes incorrect and so will work against the model achieving very low loss.

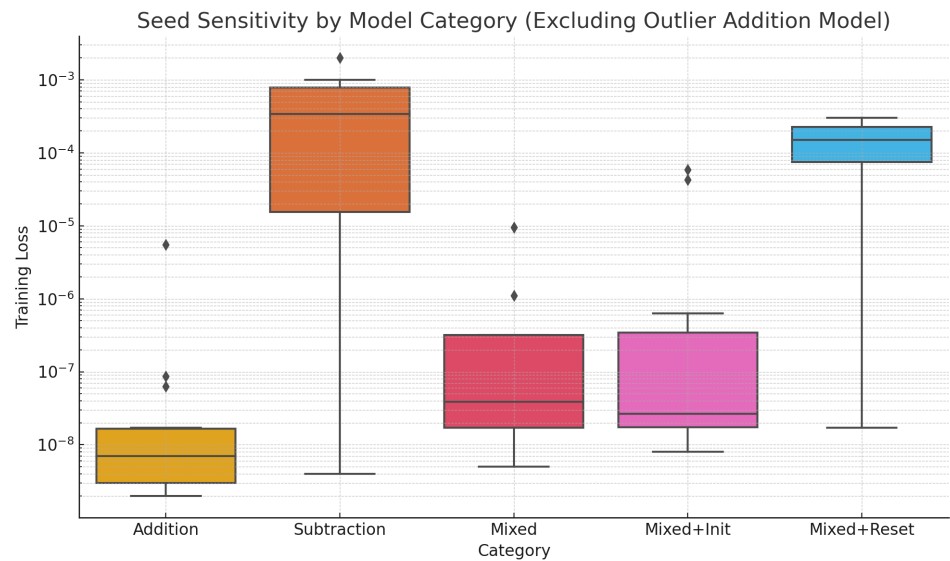

Figure 15: A visualization of the range of training losses across 48 models grouped by the different model categories.

## N APPENDIX: ADDITION HYPOTHESIS 3

The hypothesis 3 pseudo-code was derived iteratively by obtaining experimental results and mapping them to mathematical operations. Some of the experiments and mappings were:

- Ablation experiments show that the A5 value is **accurately** calculated in prediction step 11 using 5 attention heads and 5 MLP layers. The pseudo-code accurately calculates A5 while constraining itself to this many steps.

- Ablating the nodes one by one shows which answer digit(s) are reliant on each node (Ref Tab. 7). Most interestingly, ablating P10.L0.H1 impacts the answer digits A5, A4, A3, A2 (but not A1 and A0). This node is used in the calculation of A5, A4, A3, A2 in prediction steps 11, 12, 13 and 14. These relationships are constraints that are all obeyed by the pseudo-code.

- The pseudo-code has 4 instances where $ST_n$ is calculated using TriCase. PCA of the corresponding nodes (P8.L0.H1, P9.L0.H1, P11.L0.H2 and P14.L0.H1) shows tri-state output for the specified $D_n$. (see Figure 7).

- The pseudo-code has 4 instances where compound functions using TriCase and TriAdd to generate tri-state outputs. PCA of the corresponding nodes (P11.L0.H1, P12.L0.H1 and P13.L0.H1) shows tri-state output for the specified $D_n$. (see Figure 7).

- Activation patching (aka interchange intervention) experiments at attention head level confirmed some aspects of the calculations.

- The pseudo code includes calculations like ST1 which it says is calculated in P9.L0.H1 **and** P9.L0.MLP. Ablation tells us both nodes are necessary. For the attention head we use the PCA results for insights. We didn't implement a similar investigative tool for the MLP layer, so in the pseudo-code we attribute the calculation of ST1 to both nodes.

- For P10.L0.H1, the attention head PCA could represent either a bi-state or tri-state output. The MLP layer at P10.L0.MLP could map the attention head output to either a bi-state or tri-state. We cannot see which. The pseudo-code shows a tri-state calculation at P10.L0.MLP, but with small alterations the pseudo-code would work with a bi-state output.

- For P15.L0.H1 the attention head PCA could represent either a bi-state or tri-state output. The pseudo-code shows a bi-state calculation SC0 at P15.L0.H1, but with small alterations the pseudo-code would work with a tri-state output.

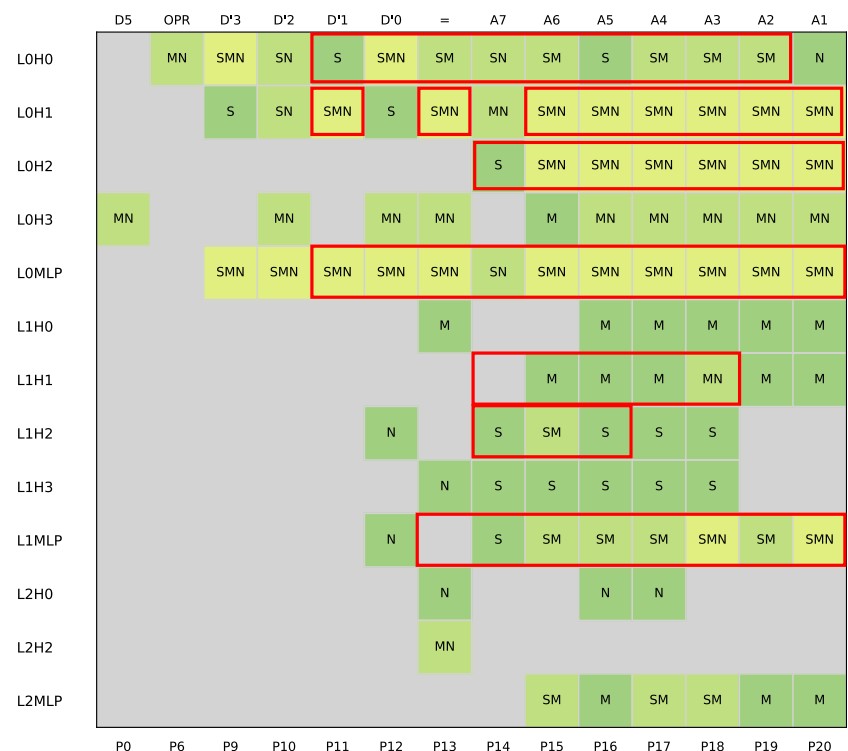

Figure 16: This map of a sample 6-digit **mixed** model shows the 98 nodes used to predict answers to addition (S), positive-answer subtraction (M) and/or negative-answer subtraction (N) questions. Before training the mixed model, 48 nodes were initalized pre-training with a smaller **addition** model's weights. These are have a red border. During mixed model training, 39 of 48 of the initalized monosemantic nodes were generalized (become poly-semantic) and now help predict two or three question classes.

- The calculation of ST2 in P14.L0.H1 is a interesting case. The model needs ST2 for A2 accuracy. The model could simply reuse the accurate ST2 value calculated in P10. Activation patching shows that it does not. Instead the P14 attention heads calculate ST1 from D1 and D'1 directly, and only relies on the P10.D1.ST2 value in the case where ST2 != ST1. That is, the calculation is "use P14.ST1 value else use ST2 value". This aligns with the model learning the P10.ST1 calculation early in training (for 90% accuracy) and later learning that P10.ST2 contains additional information it can use to get to >99.999% accuracy.

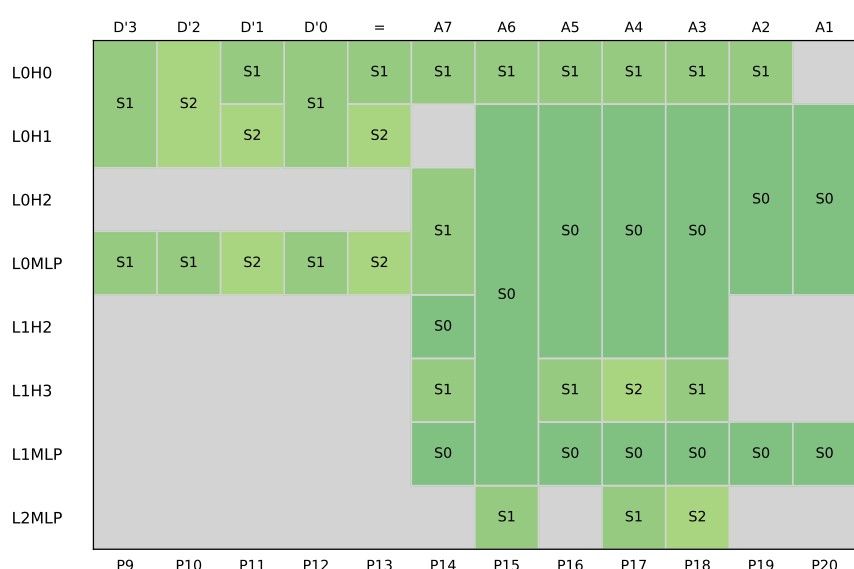

Figure 17: This map shows the simplest **complexity** quanta S0, S1, etc used in each useful node of the **6-digit** 3-layer 4-head **mixed** model when doing **addition** questions.

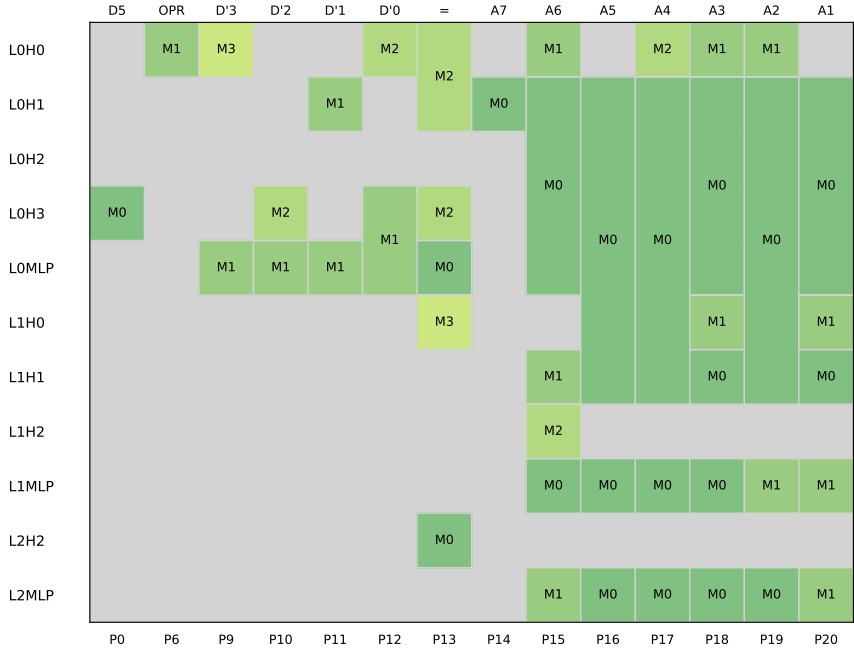

Figure 18: This map shows the simpliest **complexity** quanta M0, M1, etc used in each useful node of the **6-digit** 3-layer 4-head **mixed** model for **subtraction** questions with positive answers.

