# OpenReview forum: "Understanding Addition and Subtraction in Transformers"
_ICLR.cc/2026/Conference — Submitted to ICLR 2026_

### Official Review · Reviewer_FknJ · 2025-10-17

**Soundness:** 3
**Presentation:** 2
**Contribution:** 2
**Rating:** 2
**Confidence:** 3

**Summary:**

This paper investigates whether small, standard Transformers (2–3 layers, 3–4 heads) learn an algorithmic solution to multi-digit addition/subtraction when trained on synthetic data enriched with cascading carry/borrow cases. The authors formalize addition via a left-to-right decomposition—per-digit sums (**SA**), tri-state carry judgements (**ST** ∈ {0,1,U}), and a prefix operator (**TriAdd**) producing definite carries (**SV**) claimed to be resolved by the “=” token—then propose an analysis framework that (i) searches concrete model “nodes” (layer/head/position residual writes), (ii) enforces behavior/ordering constraints aligned with the algorithm, and (iii) performs causal verification via activation swapping on paired inputs differing in a single subtask value. They report near-perfect accuracy on 5–12 digits across 49 models and provide a toolkit for node characterization/ablation.

**Strengths:**

- **Clarity of the target mechanism.** The SA/ST/SV (TriAdd) decomposition gives a precise, human-readable algorithm that is aligned with left-to-right decoding and can be checked against model internals.
- **Methodological step beyond visualization.** Activation swapping constitutes a causal test that can, in principle, show that specific internal variables are *used* rather than merely correlated with outputs.
- **Reusable infrastructure.** The released toolkit for node search/characterization/ablation is potentially useful for similar mechanistic studies within synthetic arithmetic.

**Weaknesses:**

1) **Limited substantive reach of several listed contributions.**
   The paper enumerates contributions as (i) an LLM “survey,” (ii) enriched datasets, (iii) 49 small models with near-perfect accuracy, (iv) exact algorithms, and (v) a reusable interpretability toolkit.
   - *(i)* The “survey” functions as a test sweep rather than a principled evaluation: it lacks a clear task taxonomy or depth that would yield new insights; as written, it reads as a breadth-only diagnostic.
   - *(ii)* The enriched datasets are straightforward to synthesize for addition/subtraction; beyond standard edge-case oversampling, there is little conceptual novelty.
   - *(iii)* Near-perfect accuracy on short n-digit arithmetic with small models is not in itself a strong contribution; many prior works can achieve this under similar setups.
   - *(iv) & (v)* The algorithmic decomposition and the causal toolkit are the genuinely interesting pieces; however, their **application breadth** and **forward significance** are narrow (synthetic digit-level arithmetic). As framed, this falls short of ICLR’s bar for impact unless the paper more clearly argues why these tools produce new, generalizable insights *within the stated scope*.

2) **Key assumptions are hard-coded yet not sufficiently justified within the paper’s logic.**
   The analysis embeds strong priors—for example, that ST/SV needed for the first answer digit are computed **before** “=”. While defensible for worst-case cascades, this is stronger than necessary for the typical cut-off cases that dominate the distribution. The paper treats such priors as constraints without fully arguing their necessity (vs. design choice) for the claims actually made.

3) **Under-specified experimental pipeline at the crux of the claims.**
   The node-search/labeling procedure governs all main results, yet crucial details are missing: attention thresholds and their sensitivity, PCA/probe criteria, ordering-check implementation, per-stage pass/fail counts, tie-breaking when multiple nodes satisfy a subtask, and cross-seed variability. Without these, it is hard to assess how general or fragile the reported “consistent” findings are.

4) **Evidential weight of PCA is limited as presented.**
   PCA plots appear as case studies. The paper does not provide population-level quantification (e.g., separability/probe metrics across *all* labeled nodes) or negative controls. Given the centrality of node labeling, this weakens the argument that the behavior-constraint stage reliably identifies the intended subtasks.

**Questions:**

1. **Necessity of positional priors.** Are the “before =” constraints logically required for your claims, or are they convenient design choices to guarantee worst-case visibility? Please justify within the paper’s framework (no need to add new tasks). If they are necessary, clarify the argument; if not, state the intended scope explicitly.

2. **Pipeline transparency where results hinge on it.** Please provide precise (in-paper or appendix) specifications for the node search and labeling: the thresholds used (and brief sensitivity notes), the ordering-check implementation, tie-breaking when multiple nodes satisfy a subtask, and per-stage pass/fail counts aggregated over models/seeds.

3. **Population-level evidence for node labels.** Beyond illustrative PCA figures, can you report an aggregate separability or probe metric for all nodes finally labeled as SA/ST/SV, together with negative controls (e.g., random or incorrect-position nodes)? This stays within your main contribution (validating the proposed mechanism) while strengthening the evidential basis.

---

> ### Author Response · Authors · 2025-11-20
>
> We thank the reviewer for their thoughtful questions and address the concerns below.
>
> **Re Limited substantive reach of several listed contributions.**
>
> Please refer to my answer to reviewer crJ9’s query “W1: On Significance.”
>
> Re the “survey” functions as a test sweep: I agree. Its main point is to demonstarte that most models are bad at addition. While this paper shows that Transformers can learn to perform natural-ordering addition accurately.
>
> **Q1: Necessity of positional priors**
>
> Consider the answer digit “1” in “206727644+793272359=1000000003” (from Table 4). Mathematics requires processing **all** question digit pairs, calculating a cascading carry one through **all** digits, before the “1” can be calculated. Only after the last question digit is seen can the first answer digit be calculated. This informs our positional constraints.
>
> Empirically, our models are highly accurate on prompts like this. With billions of possible answers, memorizing is infeasible. The Transformer architecture requires the “1” is calculated at or before the “=” token. In this example, using our algorithm **all** SVn values can only be finalized when the final question digit is processed. Also, SVn relies on STn. With our algorithm an accurate model must calculate SVn and STn by the “=” token. The SAn can be and are calculated after the “=”. Figure 1 diagrams this.
>
> We searched for SAn before the “=” and STn/SVn after the “=” and did not find any. (One of our early algorithm hypotheses assumed SAn, STn & SVn all existing before the “=”. This is mathematically feasible but no model implemented this algorithm.)
>
> A parallel version of the aboce argument applies to the answer sign token in a subtraction question.
>
> **Re Under-specified experimental pipeline at the crux of the claims.**
>
> **Re Q2: Pipeline transparency where results hinge on it.**
>
> Please refer to the answer to reviewer bU7a.
>
> Further, for the SVn, STn and SAn causal verification (activation swapping) tests, based on our expectations of the algorithm, across all our accurate models, we had a 100% success rate in predicting the model’s answer after the intervention (Lines 344-6) e.g. for a 6 digit model, in each case we correctly predicted the models altered answer out of a million possibilities.
>
> We crafted the (programmatically generated for n-digits) causal verification tests for say STn so it would not trigger on a SVn or SAn behavior. We included “negative” interventions where we expected (and saw) no change in the model’s answer.  We made public the many hundreds of lines of code that implemented this framework.
>
> **Re Population-level evidence for node labels.**
>
> Great queston.
>
> Empirically most model neurons are required for accuracy (ablating the remaining neuron does not impact accuracy.) Every neuron that is required for accuracy has exactly the characteristics associated with exactly one of the subtasks in our addition algorithm. This included the very strong per-digit ST/SV/SA-specific causal verification tests.
>
> The PCA evidence we consider indicative only. The 3 group separation differed per model and sometimes per neuron. We did not formulate any “all models” threshold.
>
> The potential counter-factual, that the models implement some different algorithm that still delivers high accuracy using those same (empirically required for accuracy) neurons, with the neurons having all the demonstrated characteristics and orderings needed for our algorithm, seems extremely unlikely.
>
> **Summary**
>
> Mechanistic Interpretability is a young science without the established proof threshold levels of more mature sciences. Our paper suffers from this lack of established norms. Nonetheless we believe the evidence we present is very robust and sufficient to establish our explanation as being (at least) very likely correct.
>
> Our paper includes practical data and training insights, novel computational algorithms, a methodology for formalizing algorithmic constraints, and then a framework for programatically searching models to search for a hypothesised algorithm.
>
> Given the above, we request that you consider a higher rating for this paper.

---

### Official Review · Reviewer_crJ9 · 2025-10-31

**Soundness:** 2
**Presentation:** 2
**Contribution:** 1
**Rating:** 4
**Confidence:** 3

**Summary:**

This paper shows that small transformers trained from scratch can perform n-digit addition and subtraction almost perfectly (99.999%).
The authors present a mathematically exact, left-to-right algorithm that uses “cascading carry one” for addition and “cascading borrow one” for subtraction, and they argue that trained transformers actually implement this procedure.
To support the claim, they train 49 tiny models with different configurations on a synthetic dataset that boosts the proportion of hard edge cases with long carry chains.
The authors also build “mixed” models that handle both addition and subtraction, showing that if we bring a module identified in a trained addition model and then train on a mixture of addition and subtraction data, the model reuses the inserted addition circuits for both tasks (become polysemantic).
Finally, the authors provide a tool for characterizing nodes that perform specific subtasks.

**Strengths:**

Unlike many prior studies that use reversed token order, this work tackles addition and subtraction in the human-familiar format.

**Weaknesses:**

**W1.**
I feel the significance of the paper is quite limited.
As I understand it, each model in the experiments receives operands of a fixed length.
If so, what is the meaning/implication of showing that a Transformer can solve fixed-length addition/subtraction with near-perfect accuracy? In practice, what we need are models that operate reliably across variable lengths and formats with many other different tasks at the same time.

**W2.**
The authors claim that the trained Transformers actually implement their proposed algorithm, but I think the evidence is not sufficiently persuasive.
Here, I reference prior papers demonstrating that trained Transformers can implement specific algorithms.
Cho et al., (2025) [1] studied length generalization for addition and proposed position coupling.
Crucially, they theoretically constructed a Transformer and proved it can solve addition; they then showed that the learned model’s attention patterns align with those of the construction, thereby clarifying which algorithm the model learned.
Similarly, though in a different setting within the in-context learning literature, Nichani et al. (2024) [2] constructed a Transformer capable of in-context learning for tasks generated by a Markov chain and again verified that the learned attention patterns align with their theoretical construction.
In short, a common approach to identifying the algorithm a trained Transformer has learned is to theoretically construct an explicit model and then demonstrate similarity with the trained one.
However, the automated testing framework proposed here (whose operation is not clearly explained in the paper) seems too weak to substantiate the strong claim that the trained Transformer implements the specific algorithm the authors describe.

**W3.**
Several definitions and explanations lack detail.
Here I list few of them.
What exactly is the $SC\_n$ task?
How does the automated testing framework operate?
In Figure 2, how many test instances did the authors use for each length?

---

[1] Cho, Hanseul, et al. "Position coupling: Improving length generalization of arithmetic transformers using task structure." NeurIPS 2024

[2] Nichani, Eshaan, Alex Damian, and Jason D. Lee. "How transformers learn causal structure with gradient descent." ICML 2024

**Questions:**

**Q1.**
What is the exact definition of a “node” in this paper?
I think the term "node" is slightly ambiguous; is it a specific weight in the Transformer, or a hidden representation at a specific token?

**Q2.**
Lines 185–186 state: “Note that the calculation first combines ST2 and ST1 before combining the result with ST0.”
If the trained Transformers truly implement the proposed algorithm, this ordering should be reflected in the learned model.
Where is the evidence supporting this claim?

**Q3.**
Suppose we wish to train on longer operands.
Among the number of layers, number of heads, and embedding dimension, which should be increased (if any)?
Or is scaling unnecessary?

---

> ### Author Response · Authors · 2025-11-20
>
> We thank the reviewer for their thoughtful questions and address each concern below.
>
> **W1: On Significance**
>
> The core contribution is not solving fixed-length arithmetic— **it is discovering and validating that transformers learn profoundly different algorithms from humans**. This advances mechanistic interpretability through:
> - **Algorithm discovery methodology** generalizable beyond arithmetic
> - **Understanding non-human computation**: The Figure 1 algorithm is mathematically correct but unintuitive. Discovering it required abandoning human preconceptions - precisely the insight needed for understanding capable AI systems. The model has learnt a "horizontally extensible" algorithm (working on 5→15 digits, with the same architecture) which can cope with the natural (highest digit first) question format.
> - **Practical insights:** Mixed-model polysemanticity (Section 4.1.1) and enriched training data recipes (Appendix D). The training data sets embed several insights gained only through a deep analysis of the models, understanding successfully rare edge cases the the models struggled to learn e.g. subtractions when the operands are identical (Lines 610-2).
>
> **W2: On Evidence Strength**
>
> We employed empirical algorithm discovery + comprehensive causal verification, complementary to the theoretical-construction approach of Cho et al. (2025). Our evidence converges from multiple independent tests:
> - **Activation patching** (lines 356-359): Swapping activations between minimal paired inputs predicts output with 100% accuracy from millions of possibilities
> - **Exhaustive constraints** (lines 342-355): Attention patterns, positioning, PCA clusters, 30+ ordering dependencies, ablation effects. All accurate models satisfy all constraints
> - **Coverage**: Majority of nodes participate; others can be ablated without impact
>
> The reviewer may underestimate evidential strength because Section 4.2.1 is concise. Page limits drove us to use english descriptions rather than pseudocode. All constraints are mathematically defined, test procedures detailed in Appendix N, results in Figures 10-14 and Tables 1-3, with the code publicly released. We are confident in the results.
>
> **W3: Clarifications**
>
> Re Automated framework: Please refer to answer to reviewer bU7a
>
> Re Figure 2 testing: Each model was tested with one carefully-chosen cascading-carry question per length (Table 4) with the score based on the longest correct answer before failure. We accept Figure 2 may overstate the ability of some models, but this only strengthens our concerns about the lack of accuracy in most models. (Appendix E updated)
>
> **Q1: What is a "node"?**
>
> In our paper a node is a logical locus implementing a specific subtask. Physically: an attention head (e.g., P18L0H1), an MLP layer (P18L0MLP), or occasionally two heads working together. We identify nodes by behavior (what they compute). Appendix A now includes this definition.
>
> **Q2: Evidence for ST2→ST1→ST0 ordering**
>
> This ordering is formalized in "Algorithmic ordering constraints" (lines 354-355). These constraints are tested programmatically by the framework - sets of candidate nodes must satisfy these ordering constraints. An anecdotal example of this ordering is visible in the Table 1 example: ST0 at P12, ST1 at P9/P11, ST2 at P6, all before "=" at P13.
>
> **Q3: Scaling to longer operands**
>
> The algorithm extends horizontally: each digit added to the prompt provides additional tokens which provide room for one more SA/ST/SV subtask to be processed. Across our training from 5 to 15 digits, we did not need architectural scaling: We used the same architecture and achieved very high accuracy. We suspect that a sufficiently long question would bottleneck on the finalization of all the SVn subtasks on the “=” token, but we did not hit this suspected limit.
>
> **Summary**
>
> We believe our contributions—particularly the novel algorithm and systematic verification across 49 models—represent a significant advance in mechanistic interpretability. We hope we have addressed your concerns. Given your assessment that the work is "good" in soundness and contribution, and your statement that you "would not mind if paper is accepted," we respectfully ask if these clarifications and revisions would justify raising your score to 5 or 6.
>
> We are happy to provide any additional details you need, and we will submit a revised PDF incorporating these improvements.

---

### Official Review · Reviewer_bU7a · 2025-11-01

**Soundness:** 3
**Presentation:** 2
**Contribution:** 3
**Rating:** 4
**Confidence:** 3

**Summary:**

This work provides several new insights into the ability of LLMs to learn multi-digit addition. While many publicly available LLMs cannot reliably perform multi-digit addition, small Transformers can learn this task when trained on datasets containing many edge cases. Unlike human computation, where calculation proceeds from the least significant to the most significant digit, Transformers predict from the most significant to the least significant digits. The authors propose a new addition algorithm that emulates this process, and careful analysis reveals that Transformers indeed implement such computational nodes. The proposed algorithm and accompanying observations elucidate the computational process inside LLMs in terms of addition, subtraction, and their combination.

**Strengths:**

- This paper tackles the addition learning task while maintaining the left-to-right digit rule of autoregressive models. Unlike many previous works that introduce tricks (e.g., zero padding, additional positional embeddings, or reversing digit order) to facilitate learning, this work keeps the learning process unchanged and aims to explain the basis for successful learning.
- The explanation is grounded in a novel addition algorithm aligned with the left-to-right digit rule. The algorithm suggests multiple subtasks, and this study discovers that the trained models contain nodes corresponding to these subtasks.
- The authors conduct extensive examinations across many pre-trained and newly trained models, and they publicly release their datasets and toolkits.

**Weaknesses:**

While I appreciate the novel attempt of this work, there are several aspects that I could not follow, partly due to issues in the presentation. I would appreciate it if the authors could clarify these points (and revise the manuscript accordingly), and I would be happy to increase my score.

In what follows, I focus on the addition case. I list my comments based on the flow of this study, rather than their importance.

---
First, while it is technically interesting to address addition learning in a left-to-right setup, the motivation is not clearly explained. Right-to-left computation is more mathematically reasonable and indeed more learning-friendly. The ordering of autoregressive generation is critical [1, 2]. The authors should explain more about the underlying motivation for focusing on a left-to-right setup and, ideally, its practical impact. It might also be interesting to compare the efficiency (or some other metric) of the proposed left-to-right addition algorithm with the standard one. Is the proposed algorithm mathematically as efficient as the standard one? If so, learning addition in a left-to-right setup should be as difficult as that in a right-to-left setup (personally, this seems doubtful).

[1] Shen et al., "Positional Description Matters for Transformers Arithmetic," 2023.

[2] Sato et al., "Chain of Thought in Order: Discovering Learning-Friendly Orders for Arithmetic," 2025.

---
Second, to my understanding, the addition model was trained without any tricks, such as additional loss. The proposed left-to-right algorithm is only used for interpretation and analysis and is not involved in the training step. The only trick used is to design the dataset to contain more edge cases. A recent study [3] reports a similar technique, training with easier samples, particularly those with many zeros in the inputs. The easy samples in [1] are also edge cases, while the reverse is not evident. This connection may suggest an interesting training recipe.

[3] Saxena+, Making Hard Problems Easier with Custom Data Distributions and Loss Regularization: A Case Study in Modular Arithmetic, ICML'25

---

Third, the verification process of the core observation—that the addition model involves nodes that correspond to the subtasks and satisfy several constraints—is unclear to me. Line 349 comments on the "automated testing framework" in three steps, but it critically lacks details. Thus, while the core observation sounds intriguing, I cannot judge its validity.

**Questions:**

Please address the weaknesses raised above. In particular, the third comment is critical and why I'm tentatively giving a negative score.

---

> ### Author Response · Authors · 2025-11-20
>
> We sincerely thank the reviewer for their thoughtful feedback and recognition of our novel algorithm and extensive experimental validation. We are encouraged by your assessment of the paper's soundness (3: good) and contribution (3: good), and we address your concerns below with concrete clarifications and proposed revisions.
>
> **On Verification Framework Details**
>
> We understand this is your primary concern and appreciate the opportunity to clarify. Section 4.2.1 provides a complete specification of our verification framework through:
> - Exhaustive constraint enumeration (lines 342-359): We list every constraint a model must satisfy to implement our algorithm:
>   - Functional accuracy (correct from millions to billions of possible answers)
>   - Complete component coverage (all subtasks for all digit positions)
>   - Component-level behavior (attention patterns, token positioning, PCA cluster structure, ablation effects)
>   - Algorithmic subtask ordering (30+ constraints on computational dependencies)
>   - Causal verification (activation swapping with exact predictions)
> - Automated testing (line 360-365): Our framework systematically:
>   - Searches for candidate nodes for each subtask
>   - Retains only nodes satisfying all constraints
>   - Compares discovered nodes against algorithmic (ordering etc) requirements
> - Extensive appendices provide implementation details:
>   - Appendix N describes specific experiments (PCA, ablation, activation patching)
>   - Tables 1-3 show concrete results for individual models
>   - Figures 10-14 visualize the discovered circuits
>
> The verification is not a black box—every constraint is testable and we report which models satisfy which constraints. The fact that our accurate models all satisfy all constraints is the key finding. The reviewer may underestimate evidential strength because Section 4.2.1 is concise. We considered adding pseudocode or algorithmic descriptions but the page constraints forced us to rely on an english description. All constraints are mathematically defined, test procedures detailed in Appendix N, results in Figures 10-14 and Tables 1-3, with the code publicly released.
>
> **On Left-to-Right Ordering Motivation**
>
> We respectfully disagree that this requires further justification—left-to-right is the natural order for LLMs trained on real-world data. All publicly available LLMs encounter arithmetic in this format throughout their training corpus. As our survey of 180 LLMs demonstrates (Figure 2), most production models fail at multi-digit addition despite being trained on trillions of tokens containing arithmetic in this exact format. This is not a theoretical curiosity but a practical failure mode of real systems.
>
> The reviewer suggests comparing efficiency with right-to-left ordering, but this misunderstands our contribution. We are not arguing that left-to-right is more efficient—we are demonstrating that:
> - Models can be trained to near-perfect accuracy in the natural left-to-right format (no tricks needed, just great training data)
> - We can reverse-engineer the learned algorithm, which proves profoundly different from both human computation and right-to-left approaches
> - This algorithm generalizes to mixed operations (addition and subtraction)
>
> The practical impact is immediate: our work shows that with enriched training data, transformers can reliably learn arithmetic in the format they actually encounter. The theoretical impact is that we've identified a novel, interpretable algorithm that emerges naturally from gradient descent.
>
> We will add sentences in the introduction explicitly stating that left-to-right is the natural order for production LLMs and clarifying that our focus is on algorithm discovery.
>
> **On Dataset Design and Prior Work**
>
> Thank you for pointing to Saxena+ (ICML'25) etc. We are aware of this and other similar works on arithmetic. Many focus on easier samples (e.g. many zeros), or context simplification (e.g. reverse ordering), whereas our approach trains on harder samples (cascading carry/borrow cases) in a natural-order setting. Our enrichment strategy specifically targets the computational bottleneck (cascading operations) that causes LLM failures, as evidenced by which examples fail in our survey (Table 4). The 60% enrichment rate we describe (Appendix D) was empirically determined to minimize loss while maintaining model compactness.
>
> **Summary**
>
> We believe our contributions—particularly the novel algorithm and systematic verification across 49 models—represent a significant advance in mechanistic interpretability. We hope we have addressed your concerns. Given your assessment that the work is "good" in soundness and contribution, and your statement that you "would not mind if paper is accepted," we respectfully ask if these clarifications and revisions would justify raising your score to 5 or 6.
>
> We are happy to provide any additional details you need, and we will submit a revised PDF incorporating these improvements.

---

### Meta-Review · Area_Chair_sWoF · 2026-01-04

**Summary:**

## Summary
This submission studies how trained transformer models perform multi-digit addition and subtraction. The authors propose new algorithms for addition and subtraction that allow one to predict answers from the most significant digit (MSD) to the least significant digit (LSD). This differs from the human procedure, which typically proceeds from the LSD to the MSD. The authors developed an automated testing framework to test if transformer models trained to solve the tasks really contain subtasks defined by the authors’ algorithms. The paper claims that trained models do implement such subroutines.


## Reviewer Concerns
Major concerns raised by reviewers can be summarized as the following.
- **Motivation/significance**. Reviewers bU7a and crJ9 questioned the importance/significance of studying transformers achieving near-perfect accuracy on fixed-length MSD-to-LSD generation of addition/subtraction solutions. Reviewer FknJ pointed out that while the algorithmic decomposition and the automated toolkit are interesting, their application breadth looks limited.
- **Under-specified verification pipeline**. All reviewers pointed out that the paper does not describe the details of the automated testing framework, which is one of the core contributions of the paper.
- **Insufficient evidence**. Some reviewers pointed out that the empirical evidence is not substantial enough to conclude that trained transformers actually implement the proposed addition/subtraction algorithms. In particular, population-level quantification was found to be missing.
- **Readability issues**. Reviewer crJ9 notes that several definitions and explanations lack detail.

**Reviewer Concerns:**

The authors provided brief responses to the reviews. Unfortunately, the discussion period closed before we get any feedback from the reviewers. Based on my reading, the authors’ responses can be summarized as
- **Motivation/significance**. The authors responded that MSD-to-LSD generation is what LLMs actually do, and clarified that the paper is about discovering and validating that transformers learn profoundly different algorithms from humans. While I agree that the work contributes to mechanistic interpretability, I remain unconvinced about the generality of the proposed framework, in particular whether it can meaningfully extend beyond tightly controlled algorithmic tasks (e.g., arithmetic) to more realistic settings such as language modeling.
- **Under-specified verification pipeline**. The authors clarified that the lack of space led them to describe the method only in plain language. However, the response to Reviewer bU7a mostly reiterates the concise descriptions and offers a few additional pointers inside the paper. Importantly, the authors did not revise the manuscript to incorporate these clarifications, leaving the core verification pipeline under-specified in the main text.
- **Insufficient evidence**. The authors pointed to several different parts of the paper to claim that “We are confident in the results” and “The potential counter-factual seems extremely unlikely”. However, in my judgment, these responses do not sufficiently address the concern that the empirical evidence is incomplete. The main body of the paper does not present any population-level result; it only verbally claims that “we find all models contain the subtasks required by our addition algorithm, and satisfy the algorithmic requirements” without presenting any quantitative or population-level evidence in the main paper.
- **Readability issues**. The response clarified some of the confusions. I would like to note that, as Reviewer crJ9 mentioned, failure to formally define the key term “node” significantly affects the clarity and interpretability of the presentation.

Overall, while the responses addressed some of the concerns, the unclear description of the core verification pipeline and the lack of concrete evidence still remains outstanding, in my view.

**Reviewer Scores:**

All initial reviews were negative, there are unresolved concerns after the rebuttal, and the authors did not make any rebuttal revision. Given this, I expect that the reviewers wouldn’t have raised the scores. Overall, the paper would require substantial clarification of its core methodology and significantly stronger empirical validation to meet the bar for acceptance. Therefore, I cannot recommend acceptance at this time.

---

### Decision · Program_Chairs · 2026-01-26

Reject